# Clustering of eruptive events from high precision strain signals recorded during the 2020-2022 lava fountains at Etna volcano (Italy)

Luigi Carleo[1], Gilda Currenti[1], Alessandro Bonaccorso[1]

[1]Istituto Nazionale di Geofisica e Vulcanologia, Sezione di Catania – Osservatorio Etneo, Catania, 95125, Italy

*Correspondence to*: Gilda Currenti (gilda.currenti@ingv.it)

**Abstract.**

Lava fountains at Etna volcano are spectacular eruptive events characterized by powerful jets that expel hot mixtures of solid particles and volcanic gases reaching easily stratospheric heights. Ash dispersal and fall-out of solid particles affect the inhabited areas, often causing hazards both to infrastructures and to air and vehicular traffic.

We focus on the extraordinary intense and frequent eruptive activity at Etna in the period December 2020 - February 2022, when more than 60 lava fountain events occurred with various ejected magma volume, lava fountain height, duration. Differences among the events are also imprinted in tiny ground deformation caught by strain signals recorded concurrently with the lava fountain events reflecting a strict relationship with their evolution. To characterize this variability, which denotes changes in the eruption style, we clustered the lava fountain events using the k-means algorithm applied on the strain signal.

A novel procedure was developed to ensure a high-quality clustering process and obtain robust results. The analysis identified four groups of strain variations which stand out for their amplitude, duration and time derivative of the signal. The temporal distribution of the clusters highlighted transition in different types of the eruptions revealing thus the importance of clustering the strain variations for monitoring the volcano activity and evaluating the associated hazards.

## 1 Introduction


In the last decade, lava fountains represented a typical eruptive style at the Etna volcano (i.e., Calvari et al., 2018; Andronico et al., 2021). These eruptive events are powerful jets that can expel hot mixtures of solid particles and volcanic gases to heights ranging from tens to several hundred meters (Fig. 1a). The ash dispersal and fall-out deposits of the solid particles, known as tephra, cause critical hazards to civil infrastructures and to aviation, frequently provoking the temporary closure of southern

Italy airports. The characterization of such eruptive events is thus fundamental for both monitoring the volcano activity and evaluating the associated hazards.

At Etna, lava fountains produce short-term and small deformations of the ground (Bonaccorso and Calvari, 2017; Bonaccorso et al., 2013b; Bonaccorso et al., 2016; Bonaccorso et al., 2021) that are well-captured by the Sacks-Evertson dilatometer (Fig. 1b; Sacks et al., 1971), a widely employed geophysical instrument to study ground deformation processes associated with

volcano unrest (i.e. Amoruso et al., 2015; Bonaccorso et al., 2012; Bonaccorso et al., 2020; Linde and Sacks, 1995; Linde et al., 2016; Linde et al., 1993; Voight et al., 2006). This dilatometer is particularly appropriate to monitor lava fountains since it measures the volumetric strain within a very wide frequency range ($10^{-7}$ to >20 Hz) and with the highest resolution ($10^{-10}$ to $10^{-11}$) achievable among geophysical instruments (i.e. NASEM, 2017; Roeloffs and Linde, 2007). Other geodetic techniques such as GPS and InSAR are unable to detect the deformations associated with lava fountains because of their lower accuracy

(GPS > 0.5 cm) or lower frequency sampling (InSAR periodic passages). These technical characteristics make the strain measurements fundamental for monitoring explosive events, especially when images from surveillance cameras do not allow the event detection because of poor visibility (Carleo et al. 2022b; Calvari and Nunnari, 2022).

From December 2020 to February 2022, Etna underwent an intense eruptive activity with more than 60 lava fountains from the South East Crater (Calvari and Nunnari, 2022). A variability in terms of duration, degree of explosiveness and portion of

effusive flows, has been observed (Calvari and Nunnari, 2022; Calvari et al., 2022), implying a different degree of the hazard associated with these eruptive events. Indeed, the onset and the dynamic of the lava fountain is usually a gradual growing process, starting from weak Strombolian activity, continuing with transitional explosive activity, and eventually leading to sustained eruptive columns. The intensity and the duration of these three main phases are not always the same and characterize the temporal evolution of the episodes. A preliminary inspection on the strain signal recorded during the lava fountains reveals

a similar pattern for all the events and a strict relationship with their temporal evolution (Bonaccorso et al., 2021; Calvari et al., 2021) allowing tracing the waxing and waning of each episode and marking the onset and the end of the eruptions. On average, but not systematically, some differences arise in terms of amplitude and duration of the strain signal. For example, the lava fountains occurring in February – April 2021 were characterized by strain changes with high amplitudes (hundreds of nstrain) and temporal evolutions ranging from tens of minutes to 8-9 hours. Conversely, the strain changes accompanying the

eruptions in May – June 2021 were lower in amplitude (tens of nstrain) and developed in intervals from 1 hour to less than 4 hours (Fig. 2).

In the recent past, attempts to classify the lava fountains at Etna have been made manually by the experts by comparing different geophysical and volcanological data. Andronico et al. (2021) manually found different eruptive styles at the Etna volcano on the basis of volcanological observations. Calvari et al. (2022) analysed three lava fountain episodes that occurred

in 2021 with a multidisciplinary approach and gave insights into the different eruptive styles. Manual classification is time consuming since it involves a huge amount of data analysis and it is prone to subjective biases. With the aim of avoiding a classification biased by experts' belief, we investigate whether an objective cluster analysis on instrumental dataset could help in discovering group of events with similar characteristics. Clustering analyses on monitoring signals have already been performed in volcanology (Cirillo et al. 2022; Corradino et al., 2021; Langer et al., 2009; Nunnari, 2021; Romano et al., 2022;

Unglert et al., 2016) but never applied on the strainmeter data for clustering eruptive events.

Here, we made use of clustering techniques applied on the strain variations recorded concurrently with the eruptive episodes from December 2020 to February 2022 in order to derive the key features that characterize the eruptive process and distinguish the events. In particular, we applied the k-means clustering algorithm, a widely employed unsupervised machine learning

technique to solve clustering problems in several domains (Lloyd, 1982; MacQueen, 1967). One of the drawbacks of such

algorithm is that the optimal number of the clusters and also the optimal set of key features which lead to a high-quality

clustering are not known *a priori*. We developed a procedure to appropriately identify the features and the number of clusters

which ensure high cohesion and separation. Moreover, since the clustering solution could depend on the initial position of the

barycentre of the clusters (centroids) chosen to start the algorithm (Fränti and Sieranoja, 2019), we also investigated the

influence of the initial position of the centroids on the k-means performance by comparing different initialization techniques.

Lastly, we discuss the implications that this result entails in the assessment of volcanic activity and the associated eruptive

style.

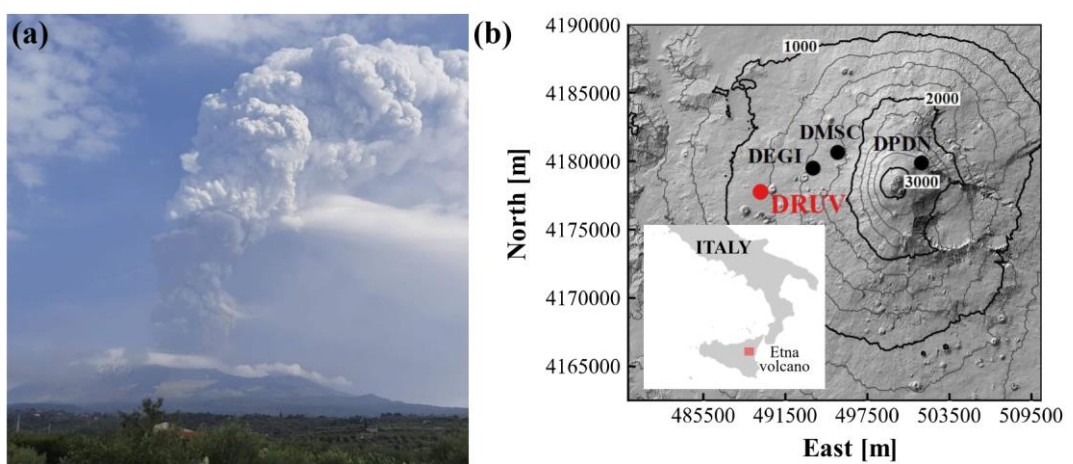

**Figure 1: (a) Lava fountain occurring on 23 October 2021 (photo from INGV internal report n° 43/2021). (b) Location of the borehole**

**strainmeter stations installed at the Etna volcano. The coordinates system is WGS 84 UTM 33S.**

## 2 Strain changes during the Etna lava fountains in 2020-2022

The December 2020 - February 2022 Etna eruptive activity was extraordinarily intense. It started with four lava fountains from

13 December 2020 to 16 January 2021 (period $P_i$). Successively, a first lava fountain sequence of 17 events took place up to

1 April 2021, with an average frequency of 0.39 events/day (sequence $S_1$). After 49 days of repose, Etna volcano reawakened

and a second lava fountain sequence of 34 episodes occurred till 10 August 2021, with a frequency of about 0.42 events/day

($S_2$). Then, the eruptive activity diminished with 5 events occurring from 10 August 2021 to the end of February 2022 ($P_f$).

The borehole strainmeter network, operating at Etna since 2011 (Bonaccorso et al., 2016; Fig. 1b), was fundamental in

investigating the dynamics of the eruptions (Bonaccorso et al., 2021) and monitoring the eruptive events in near real-time for

volcanic surveillance (Carleo et al., 2022b). In this study, we focus on the measurements recorded by the DRUV station, which

is located quite far from the summit craters, at ~11 km, and installed in a massive rock layer (at ~180 m depth) guaranteeing

high-efficiency in transferring deformation from the rock to the sensor. The strainmeter was calibrated with three different

techniques (Bonaccorso et al., 2013a; Bonaccorso et al., 2016; Currenti et al., 2017) that confirmed its high sensitivity (~10⁻

[10]). The DRUV strain signal was filtered from the disturbing effects of both the Earth tides and the barometric pressure to highlight small strain variations related to the volcano activity (Currenti and Bonaccorso, 2019). We used the procedure proposed by Carleo et al. (2022a) to highlight tiny volcano-related strain changes up to $10^{-10}$ for time scales less than 1 day. Furthermore, we removed the long-term drift component from the strain signal due to the effect of both the curing of the cement and the relaxation of the drilled hole (Canitano et al., 2021).

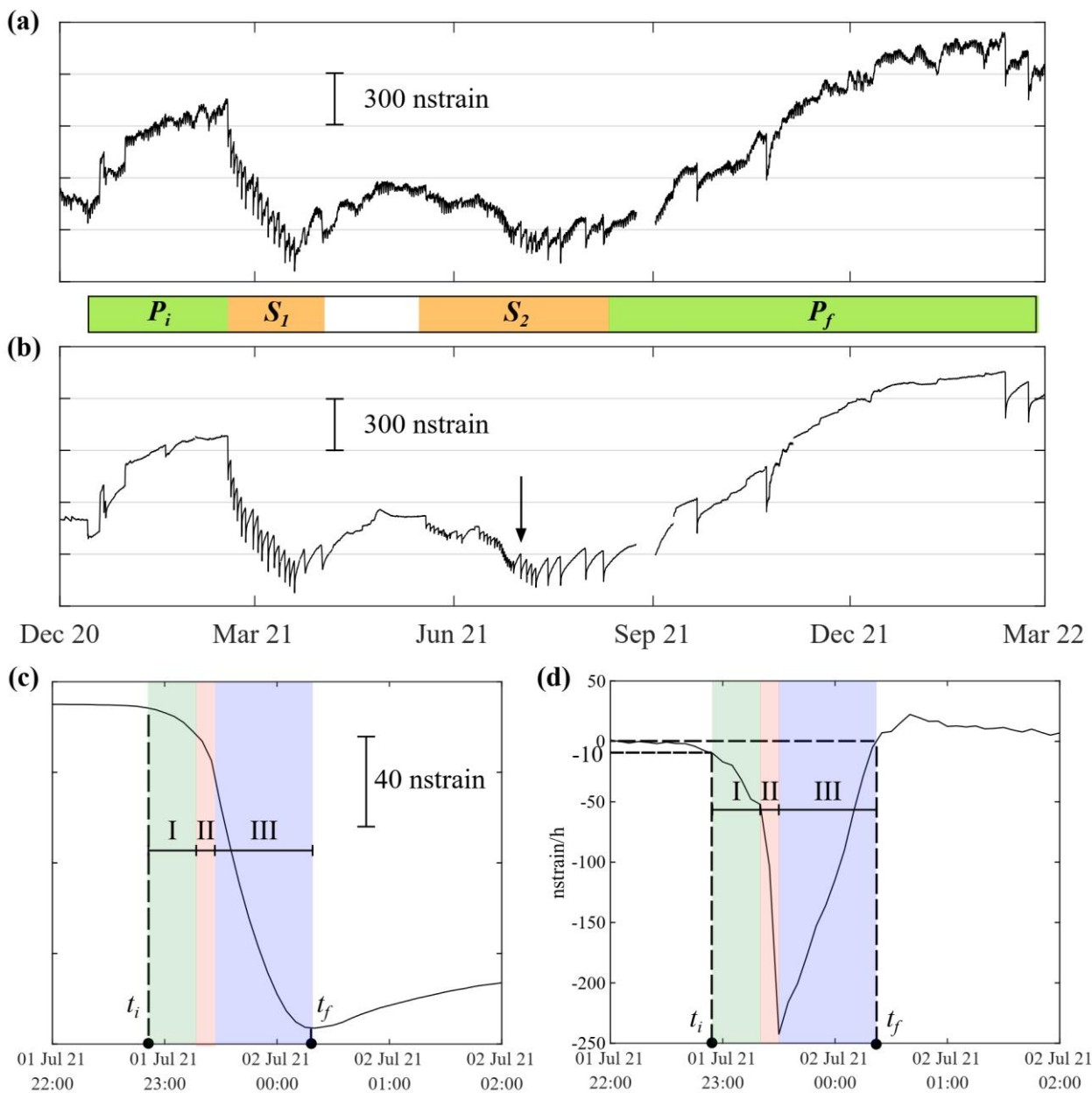

**Figure 2: The detrended recorded (a) and filtered (b) DRUV signals in the study period. Negative step-like strain variations, clearly detected in the filtered signal, occurred concurrently with the lava fountain episodes. The lava fountains on 29 August and 23 October 2021 were not recorded due to technical problems. The eruptive activity is divided into 4 periods: $P_i$ (4 events), $S_1$ (17 events), $S_2$ (34 events) and $P_f$ (5 events). The black arrow in (b) indicates the lava fountain events occurred on 01 July – 02 July 2021. The zoom of the the filtered strain signal (c) and the related strain rate signal (d) during such event are shown as an example. $t_i$ and $t_f$ indicate**

**the onset and the end of the strain variations. At the beginning of the lava fountain (I), expansion rate at the sensor gradually increases. In the central part (II), strain rate rapidly reaches its absolute maximum value. The expansion gradually diminishes up to a minimum and the strain rate reaches the pre-event level during the final part of the eruptive episode (III).**

In Fig. 2a and 2b, both the recorded and the filtered DRUV signals are presented for the period December 2020 – February

2022. The near-real time detection algorithm developed by Carleo et al. (2022b) allowed the automatic detection of 58 strain variations, all related to the eruptive episodes in the analysed period. Such strain variations are the response to the decompression of the magmatic source feeding the lava fountain (Bonaccorso et al., 2013; Bonaccorso et al., 2016; Bonaccorso and Calvari, 2017; Bonaccorso et al., 2021; Currenti and Bonaccorso 2019). The time derivative of the filtered strain signal (strain rate signal), like other high-precision geodetic signals (Kozono et al, 2013; Ichihara, 2016), is expected to be related to

the rate of magma chamber decompression and, thus to the speed of magma ascent (Hreinsdóttir et al., 2014). The Etna lava fountains grow gradually starting from Strombolian activity and evolving towards a sustained eruptive column. As already found in previous studies (Calvari et al., 2021; Calvari et al., 2022), the evolution of a lava fountain is well represented by both the strain and the strain rate signals. In Fig. 2c and 2d, the filtered strain and the filtered strain rate signals during the lava fountain on 1 – 2 July 2021 are shown as an example of the recorded variations.  Typically, the strain and the strain rate signals

show a sigmoid and a V shape, respectively. The different lava fountain phases can be described by dividing the signals in three main parts: in the initial part (Part I), when the Strombolian activity takes place, both the strain and the strain rate gradually decrease with time showing an elbow with a downward concavity; in the central part (Part II), the lava fountaining is persistent and the strain rate changes its slope abruptly reaching the absolute maximum value; in the final part (Part III), the eruptive activity starts declining and the strain rate inverts its trend reaching the pre-event level. To identify the beginning of

the event, we focused on the strain rate signal. We first evaluated the amplitude of the background noise of the strain rate signal, $\sigma$, as the mean standard deviation in a moving time window of 3 hours. We found a value of $\sigma$ of 0.93 nstrain/h. The beginning of the variation $t_i$ was chosen concurrently with the time when the beginning of the deformation rate can be clearly identified, namely when the strain rate exhibits a value of one order of magnitude higher than $\sigma$. Therefore, we selected $t_i$ as the time when the strain rate reaches -10 nstrain/h. The end of the variation $t_f$ was set when the sign of the strain rate becomes

positive.

We characterized each lava fountain event by extracting the main features from both the strain and strain rate signals in the period $t_i$ and $t_f$. In particular, we focused on Parts I and II of the signals. The extracted features transform each eruptive event into a strain data point in the feature domain which forms a dataset that is going to be clustered.

## 3 Clustering the strain variations with the k-means algorithm

The k-means is an unsupervised machine learning algorithm (Lloyd, 1982; MacQueen, 1967) designed to partition data points into clusters by minimizing the sum of the squared distances (SSE) between every data point and its nearest cluster mean (centroid). The data points are formed by a set of features which should be chosen by following two rules: the features should identify the data point uniquely and the smaller the feature vector, the better (Langer et al., 2009).

The k-means algorithm starts by selecting the initial centroid position in the feature domain. Each data point is assigned to the

$k$-th cluster represented by the closest centroid to the data point. The initial positions of the centroids, which represent the barycentre of the related clusters, change on the basis of the assigned data points. Iteratively, new centroid positions are re-estimated and the data point are eventually re-assigned to the closest cluster. The algorithm stops until centroids no longer move. The k-means algorithm has excellent fine-tuning capabilities (Fränti and Sieranoja, 2019). However, the goodness of the clustering solution depends on the choice of both the number of cluster $k$ and the set of feature $C$ to cluster the data.

Moreover, the results are influenced by the centroid positions chosen to start the algorithm.

To overcome the drawbacks of k-means, we tried different clustering solutions by varying the inputs of the algorithm, namely the number of clusters, the employed set of features and the initial centroid positions. The quality of the different clustering solutions was estimated by employing two methods: the "Elbow method" and the Silhouette value (*Sil*; Kaufman and Rousseeuw, 2009). The Elbow method is one of the most widely used techniques to find the optimal number of clusters. It is

a method based on the SSE value: the less the SSE of the clustering solution, the better the clustering. Normally, the higher the number of the clusters, the lower the SSE. In a $k$ - SSE plot, the elbow of the curve represents the optimal number of clusters for the analysed dataset and corresponds to the most effective clustering solution in terms of $k$ and SSE.

The Silhouette value (*Sil*) for a single data point is defined as:

$$Sil = \frac{b-a}{max\{b,a\}} , \qquad (1)$$


where $b$ is the average distance between the datum and the data of another cluster minimized over the clusters, and $a$ is the average distance between the datum and the data within the cluster to which the datum belongs. The *Sil* value is a measure of how much a data point is cohesive within its own cluster (distance $a$) and, at the same time, separated from the other clusters (distance $b$). It ranges from -1 to 1, where -1 corresponds to a completely wrong clustering while 1 to an optimal clustering.

We designed an iterative procedure to find the optimal number of cluster, $k_{opt}$, and the optimal set of features, $C_{opt}$, that allow for a high quality clustering solution for our dataset of strain variations. We analysed different clustering solutions $(C, k)$ by varying the number of clusters $k$ and the involved subset of feature $C$ and evaluated the quality of the clustering by using both the Elbow method and the Silhouette value. The initial centroid position was chosen randomly. To have a more robust result, for each analysed clustering solution, we performed $n = 10^4$ repetitions of the k-means algorithm, setting different random

initial seeds and keeping fixed the other inputs. Then, we chose the solution with the lowest SSE value. The robustness of the choice of $n$ and of the initial random position of the centroids in providing reliable results will be also proven.

The sets of the features used in the iterations of the procedure are extracted from a set of 15 potential features $X = \{X_1, X_2, \ldots, X_j\}$ (Table 1), where $j$ represents the $j$-th feature of $X$, which were taken into account to describe the strain variations in the part I and II of both the strain and the strain rate signals. Since the features are in different units and ranges, we normalized them in the range [0 1] to ensure a balanced weight in the clustering process (Langer et al., 2020). The procedure is organized in the following steps:

1) create the most basic subset of features $C_{start}$ composed by the amplitude $A$ and the duration $D$ of the strain variation;

2) if it is the first iteration of the procedure, the starting subset of feature related to the $i$-th iteration, $C_i$, is $C_{start}$ otherwise $C_{i-1,j\_max,}$ defined at point 7);

3) consider a new set of features $X_{left} = X - C_i$. Create all the possible subsets of features, $C_{i,j}$, composed by $C_i$ plus one feature from $X_{left}$;

4) cluster the dataset using $C_i$ and all the $C_{i,j}$;

5) find the optimal number of clusters for the $i$-th iteration of the procedure, $k_{opt,i}$, by comparing all the $k - SSE$ curves;

6) at $k_{opt,i}$, calculate the Silhouette values averaged over the clusters related to $C_i$ and all the $C_{i,j}$, $Sil_{a,Ci}$ and $Sil_{a,Ci,j}$ respectively;

**Table 1: Features considered in the cluster analysis**

| Symbol | Description |
| --- | --- |
| $A$ | Amplitude of the strain change from $t_i$ to $t_f$ |
| $D$ | Duration of the strain change from $t_i$ to $t_f$ |
| $Sr_{min}$ | Minimum strain rate from $t_i$ to $t_f$ |
| $SA_{0-75}$ | Amplitude from $t_i$ to the instant when 75% of $Sr_{min}$ is reached |
| $SA_{0-100}$ | Amplitude from $t_i$ to the instant when 100% of $Sr_{min}$ is reached |
| $SD_{0-75}$ | Length of the time window from $t_i$ to the instant when 75% of $Sr_{min}$ is reached |
| $SD_{0-100}$ | Length of the time window from $t_i$ to the instant when 100% of $Sr_{min}$ is reached |
| $SS_{0-75}$ | Average strain rate from $t_i$ to the instant when 75% of $Sr_{min}$ is reached |
| $SS_{0-100}$ | Average strain rate from $t_i$ to the instant when 100% of $Sr_{min}$ is reached |
| $AS_{0-50}$ | Average strain rate from $t_i$ to the instant when 50% of $A$ is reached |
| $AS_{0-75}$ | Average strain rate from $t_i$ to the instant when 75% of $A$ is reached |
| $ASr_{min,0-50}$ | Minimum strain rate from $t_i$ to the instant when 50% of $A$ is reached |
| $ASr_{min,0-75}$ | Minimum strain rate from $t_i$ to the instant when 75% of $A$ is reached |
| $AD_{0-50}$ | Length of the time window from $t_i$ to the instant when 50% of $A$ is reached |
| $AD_{0-75}$ | Length of the time window from $t_i$ to the instant when 75% of $A$ is reached |

7) if max$\{Sil_{a,Ci,j}\} > Sil_{a,Ci}$ then define a new subset of features $C_{i,j\_max}$ composed by $C_i$ plus the feature that provides max$\{Sil_{a,Ci,j}\}$; repeat from point 2) to point 7) updating $C_i$ with $C_{i,j\_max}$. If max$\{Sil_{a,Ci,j}\} \leq Sil_{a,Ci}$, stop the procedure and take $C_i$ as the optimal set of features, $C_{opt}$, and $k_{opt,i}$ as the optimal number of clusters, $k_{opt}$.

The influence of the initialization was investigated by comparing two seeding techniques: the random centroid (RC) position (MacQueen, 1967) and the method proposed by Yelda et al. (2010) (YC). The former method is the most popular and consists of locating the centroids randomly in the range of variation of the features, namely, in our case, between 0 and 1. The latter method involves first sorting data points in accordance with their distance from the origin and, then, partitioning them in $k$ clusters with equal number of sorted points. Yelda et al. (2010) proposed to locate the initial centroid position in the barycentre of each cluster. We introduced more randomness by locating the centroids randomly in each cluster. The tests were performed using the optimal set of features $C_{opt}$ found with the iterative procedure previously described. To investigate the importance of performing repetitions of the k-means algorithm choosing different initial centroid positions, we repeated the algorithm $n$ times, with $n$ in the range [10 $10^6$].

## 4 Clustering results

We used the k-means algorithm to characterize the 2020-2022 lava fountain events using the associated strain changes. analyse The iterative procedure provided the optimal number of clusters, $k_{opt}$, and the optimal subset of features, $C_{opt}$, that allow for a high quality clustering of the strain changes. The procedure converged in two steps whose results are presented in Fig. 3. Fig 3a shows the $k - SSE$ curves related to all the subsets of features analysed in the last step of the procedure. It can be seen that the elbow of most of the curves is at $k = 4$ which can thus be selected as the optimal number of clusters, $k_{opt}$, for our dataset of strain variations. By exploring the Silhouette values of all the analysed clustering solution at $k_{opt}$, the optimal subset of features $C_{opt}$ is selected in correspondence of the maximum $Sil$ value. The optimal subset is composed by three elements: (i) the amplitude $A$ and (ii) the duration $D$ of the strain variation and (iii) the average strain rate in the time window ranging from $t_i$ to the time when the strain rate reaches the 75% of the minimum strain rate, $SS_{0-75}$. The $Sil$ value of the ($C_{opt}$, $k_{opt}$) solution, obtained averaging among all the single $Sil$ values associated with the clustered data points, is very high and equal to 0.83 confirming the goodness of the clustering. Fig. 3b shows the single $Sil$ values, presented on the $x$-axis, of all the data points grouped in the related cluster indicated by the $y$-axis, for the optimal clustering solution ($C_{opt}$, $k_{opt}$), where $k_{opt} = 4$.

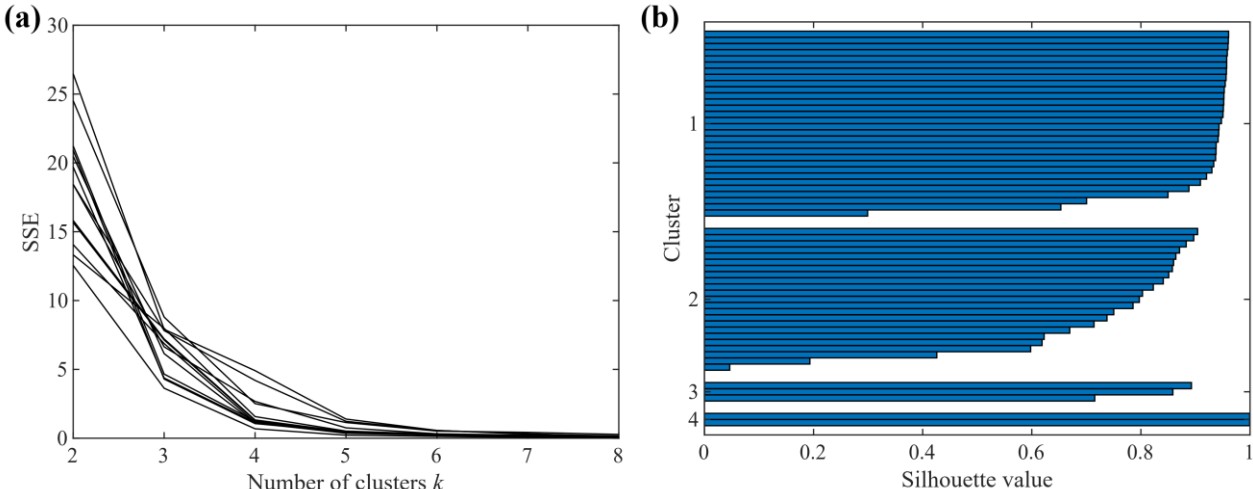

205    **Figure 3: (a)** $k$ **– SSE curves related to all the clustering solutions** $(C, k)$ **analysed in the last iteration when the procedure converged.** **(b) Silhouette values related to the optimal clustering solution** $(C_{opt}, k_{opt})$**, where** $k_{opt} = 4$**, for all the data points grouped in the related cluster indicated by the** *y***-axis.**

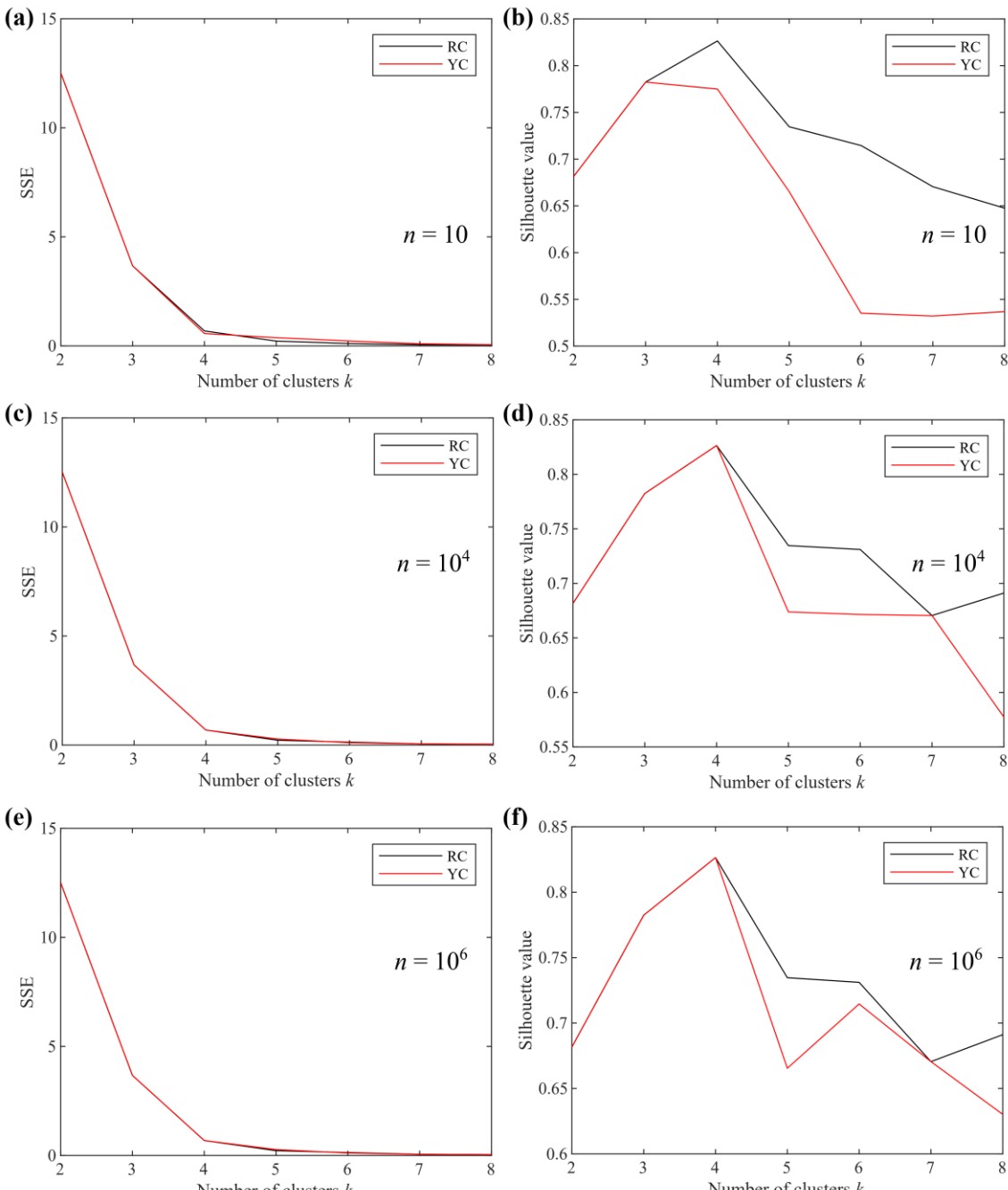

Figure 4: $k$ – SSE and $k$ – $Sil$ plots obtained using the optimal set of features $C_{opt}$. The SSE and Silhouette values are presented considering the random initial centroid (RC) positions and centroids located as proposed by Yelda et al. (2010) (YC). The results obtained performing a number of repetitions $n$ equal to 10 (a and b), $10^4$ (c and d) and $10^6$ (e and f) are shown as an example.

**Table 2: Coordinates of the cluster centroids and mode values of the optimal features for each cluster.**

| Clusters | Amplitude $A$ [nstrain] | | Duration $D$ [hour] | | Strain rate $SS_{0-75}$ [nstrain/h] | |
|---|---|---|---|---|---|---|
| | Centroid coordinate | Mode | Centroid coordinate | Mode | Centroid coordinate | Mode |
| **Cluster 1** | 50.0 | 51.5 | 1.15 | 0.98 | 32.5 | 20.4 |
| **Cluster 2** | 177.5 | 181.8 | 1.92 | 1.78 | 59.1 | 40.5 |
| **Cluster 3** | 232.7 | 225.3 | 0.83 | 0.98 | 187.6 | 181.4 |
| **Cluster 4** | 116.8 | 116.7 | 9.25 | 8.94 | 12.3 | 20.4 |

The *Sil* values are all positives indicating a good clustering for all the strain data points. Moreover, the Silhouette values averaged among the points within the same cluster are very high and equal to 0.90, 0.71, 0.82 and 0.99 for Cluster 1, Cluster 2, Cluster 3 and Cluster 4, respectively. These values denote both a high cohesion in the same cluster and a high separation among the clusters.

The results of the analysis on the influence of the initial centroid position and on the number of repetitions of the k-means algorithm are summarized in Fig. 4. We reported the $k - SSE$ and the $k - Sil$ plots related to the optimal subset of feature $C_{opt}$ and initialized with the random centroid position (RC) and the YC method for a number of repetitions, $n$, equal to 10, $10^4$ and $10^6$. The $k - SSE$ plots for the different values of $n$ (Fig. 4a,c,e) showed notably overlapped curves, indicating that the analysed initial centroid positions do not affect the shape of the curves and, hence, the choice of $k_{opt}$. In Fig. 4b,d,f the number of cluster $k$ is plotted against the *Sil* value for $n$ equal to 10, $10^4$ and $10^6$, respectively. The figures highlight that a high number of repetitions is necessary to make the clustering independent from the analysed initialization techniques. Indeed, the $k - Sil$ curves overlap only with $n$ values higher than $10^4$ and up to $k = 4$ which corresponds to $k_{opt}$. Therefore, the outputs of the procedure, $k_{opt}$ and $C_{opt}$, obtained with $n = 10^4$, can be considered reliable.

In Fig. 5a, the strain changes are presented in the $C_{opt}$ feature domain, where a very good clustering can also be observed visually confirming the reliability of the procedure in providing high-quality results. The frequency distribution of the values of the $C_{opt}$ features, $A$, $D$ and $SS_{0-75}$, are presented in Fig. 5b-d, respectively. The mode values of the frequency distributions of the $C_{opt}$ features are presented together with their centroids locations in Table 2. The analysis of the distributions of the cluster features allows us to identify the main characteristics of the events. Cluster 1 gathers lava fountain episodes of low strain amplitude and duration and characterized by small initial strain rate changes. All the features of Cluster 1 are located in the lower range of variations. Cluster 2 groups events whose features cover more the intermediate part of their ranges. Cluster 3 gathers events characterized by high deformations evolving in a very short time window, less than 1 hour. Furthermore, the mode value of the $SS_{0-75}$ feature for Cluster 3 (181.4 nstrain/h; Table 1) is 4.5 to 9 times higher than for the others. Cluster 4 groups the episodes with the highest mode value for the duration feature $D$, which is 5 to 11 times higher with respect to the ones related to the other clusters. The mode of the $SS_{0-75}$ feature in the Cluster 4 shows the lowest value among the clusters. In

Fig. 5e, all the strain changes are plotted by aligning them with their initial time $t_i$ for a further visual comparison among the clustered variations.

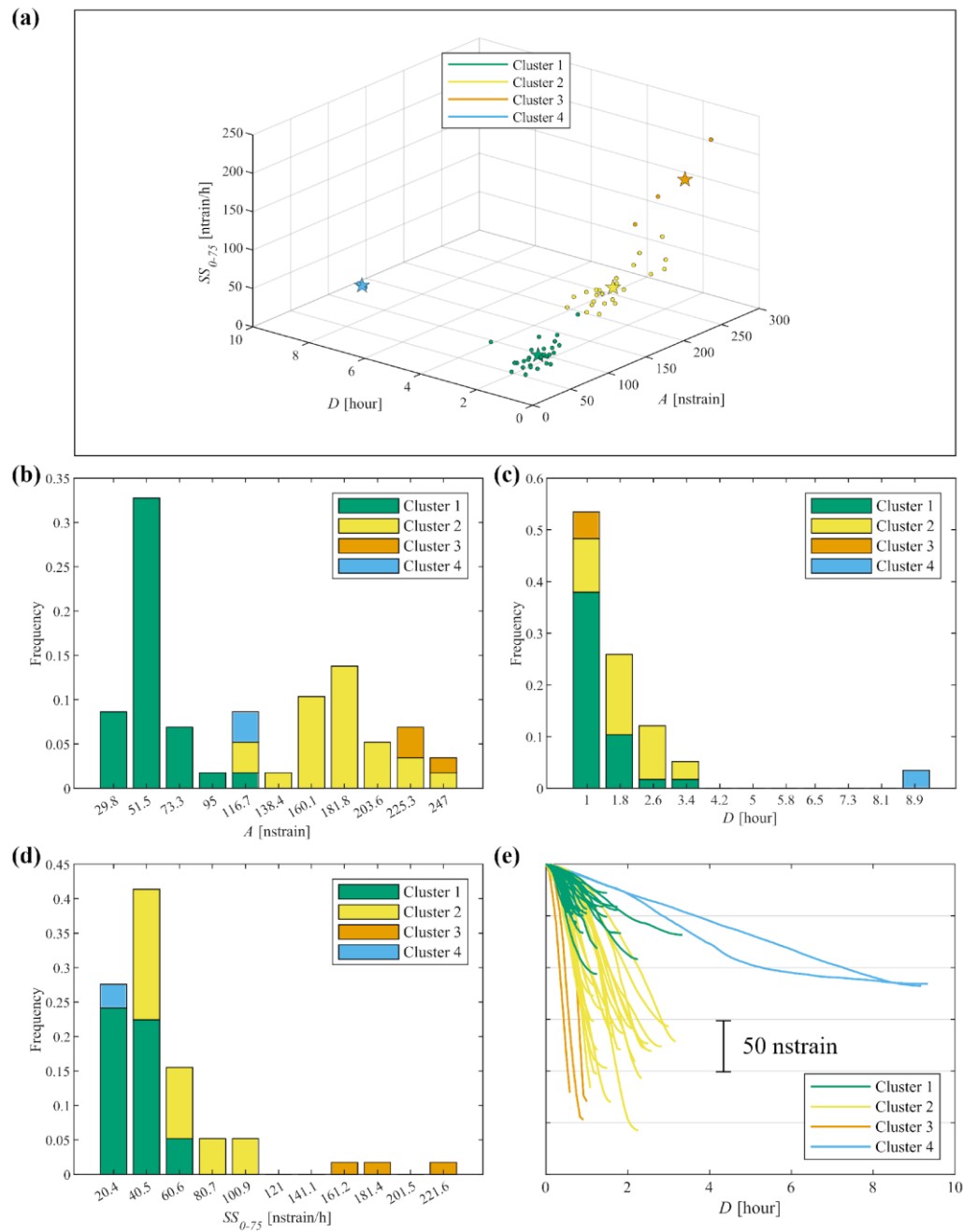

Figure 5: (a) All the clustered strain data points plotted as circles in the domain of the optimal features: amplitude $A$, duration $D$ and strain rate $SS_{0-75}$. Stars represent the cluster centroids. Frequency distribution of $A$ (b), $D$ (c) and $SS_{0-75}$ (d) for the different

clusters. Mode values of the distributions are reported in Table 1. In (e), all the clustered strain variations are aligned with the initial time $t_i$.

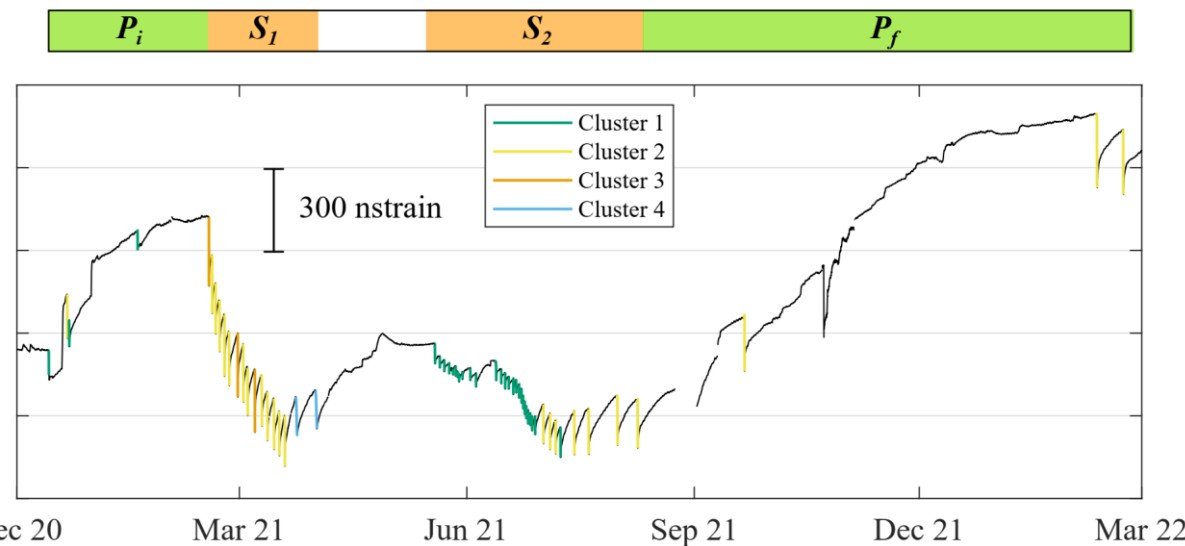

250 Figure 6: Temporal distribution of the clusters in the analysed time period plotted together with the filtered DRUV signal.

## 5 Discussions and Conclusion

For the first time, an automated clustering analysis was applied on strainmeter data to provide an objective quantitative measure of similarities and differences between explosive eruptive episodes. In particular, we studied the lava fountain events that occurred at Etna in the period December 2020 - February 2022. The number of lava fountains recognized by different
255 volcanologists in the studied period may slightly vary. However, the discrepancy in the counting of the events is due to very few weak events whose classification in a proper category was not simple for the experts themselves. Moreover, when the eruptive activity undergoes several phases of waning and waxing, often close-in-time events could be counted separately or as one (Calvari et al., 2022; Andronico et al., 2021). Despite these slight discrepancies among the experts' evaluations, at most the total number of lava fountains in the analysed period is 66 (Calvari et al. 2022). We used the protocol proposed by Carleo
260 et al. (2022b) to automatically identify the eruptive events from the filtered strain signal. By testing the protocol on the long period from 20 November 2011 to 31 March 2021, Carleo et al. (2022b) obtained a true positive rate close to 1 which means that for each lava fountain event a strain change is associated. Thanks to this high ratio we can discern, select and study the signals recorded concurrently with almost all the different explosive events. Using this protocol in the period December 2020-February 2022, we recognized 58 lava fountain events from the strain signal. Out of the 8 not recognized events, 2 were not

recorded by the DRUV strainmeter because of malfunctioning of the station and the other 6 are or very weak or counted as sub-events.

In the studied period, the eruptive events showed a high variability in their characteristics (Andronico et al., 2021; Calvari and Nunnari, 2022; Calvari et al., 2022) that is also noticed in the strain variations. Using the extraordinary 2020-2022 strain dataset, we investigated the use of an automated clustering analysis to provide an objective quantitative measure of similarities and differences of the lava fountain events.

The clustering analysis allowed us to methodically identify three key features ($A$, $D$ and $SS_{0-75}$) that grouped the events in four distinct and coherent clusters. In particular, all the three features are required to distinguish Cluster 1 and 2 from the other clusters, while $SS_{0-75}$ and $D$ sharply identify Cluster 3 and Cluster 4, respectively.

The clustered events do not occur randomly but are grouped over time as shown in Fig. 6, denoting a transition in the eruptive dynamic. It turns out that the clusters have an intimate relationship with the volcanic eruption style. In the period $P_i+S_1$ (December 2020 – March 2021), Andronico et al. (2021) manually identified three eruption styles classified as transitional activity (TA), sustained lava fountain (LF) and large-scale lava fountain (LSLF). Comparing the strain clustering and the eruption style classification reported in Andronico et al. (2021), we observed an interesting correspondence. The first events recorded in $P_i$ and classified as TA are all grouped in Cluster 1, except the 21 December lava fountain which falls into Cluster 2. Then, in the $S_1$ lava fountain sequence, the eruptive style turned into LF with episodic LSLF events. As well, the clustering highlights a transition, grouping the $S_1$ events in Cluster 2 and 3. The three events, that belong to Cluster 3, are all classified as LSLF in Andronico et al. (2021) and occurred closely in time on 16 February, 28 February and 7 March 2021. At the end of $S_1$ on 23 March and 31 March, two LF events occurred that the k-means algorithm does not associate to the Cluster 2 and requires the further Cluster 4, well separated from Cluster 2. After a period of repose, the new sequence $S_2$ restarted in May 2021 with events belonging to Cluster 1 that slowly over time turned into events belonging to Cluster 2. This transition is in agreement with a variation in the parameters estimated from the thermal camera images (Calvari and Nunnari, 2022).The comparison between the manual classification and the automatic clustering highlights that the strain signal is able to recognize and identify four classes of lava fountains, of which three are closely linked to the manual classification and a further one defining a distinctive class . In particular: Cluster 1 groups events that induce small deformation of the volcano edifice; Cluster 2 includes lava fountains to which, on average, higher deformation, higher strain rate and higher duration, with respect to Cluster 1, can be associated; Cluster 3 groups fast events (duration less than 1 hour) with high strain rate. The Cluster 4 identifies two events well separated from the others since they were characterized by very long duration and very low rate values. The peculiarity of the events in Cluster 4 was also noticed in previous studies. Calvari and Nunnari (2022) analysed thermal camera images and estimated the duration of all the 2020-2022 episodes with three different approaches. A close inspection of their results show that the duration of the two events of Cluster 4 exhibits the largest values.  Andronico et al. (2021) retrieved seismic parameters from volcanic tremor signals recorded during the eruptive episodes. The parameters related to the events of Cluster 4 show values higher than the average value estimated for the lava fountain events of the studied period.

The identification of clusters of lava fountain events has a strong impact in the alert system in place to manage volcanic crises.

During lava fountains which emit huge amounts of tephra in the atmosphere, knowledge on intensity and duration of the events has important implication, especially for civil aviation. The distinctive features of the clusters could be attributed to the degree of explosiveness and portion of effusive flows accompanying the event, that define the eruptive style. Changes in the eruptive style is regulated by many interrelated magmatic properties and processes (Cassidy et al., 2018). The exsolved and dissolved gas content, overpressure at depth, magma composition and conduit geometry of the plumbing system, all control the speed at

which magma ascends, decompresses and outgasses, and hence, determine eruptive style and evolution.

Our findings demonstrate that the clustering analysis on the strain signal may contribute in characterizing the different eruptive styles at Etna volcano and in highlighting persistence and transition in the eruptive style providing indirect insights into the evolution of the magmatic plumbing system. The obtained results are very promising and encourage us to extend it to investigate other volcanic processes that engender strain changes such as magmatic recharges and intrusions. A joint analysis,

together with other geophysical, geochemical, volcanological and petrophysical data, may help in confirming the evolution of the magmatic system conditions and in identifying the most likely magmatic properties and/or processes that regulate the volcano activity at Etna.

**Code and data availability**

MATLAB scripts and data are available upon request to the corresponding Author.

**Author contribution**

GC and LC conceived and conceptualized the study. LC developed the code and performed the analyses. AB managed and administrated the funding acquisition for conducting the research. All Authors contributed to the writing of the manuscript and the discussion of the results.

**Competing interests**

The authors declare that they have no conflict of interests.

**Acknowledgements**

This research benefited from funding provided by the 2019–2021 Agreement between INGV and Italian Presidenza del Consiglio dei Ministri, Dipartimento della Protezione Civile (DPC), All. B2 - WP2 - Task 9 and also from the EC H2020- FET OPEN project "SiC nano for picoGeo", grant agreement No. 863220. We thank the reviewers and the Editor for their helpful

and constructive comments and suggestions.

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
