# Peer review of "Clustering of eruptive events from high precision strain signals recorded during the 2020-2022 lava fountains at Etna volcano (Italy)"

_EGUsphere, 2023_

## Referee Comment (RC1)

Review of the manuscript entitled « Clustering of eruptive events from high precision strain signals recorded during the 2020-2022 lava fountains at Etna volcano (Italy) » by L. Carleo, G. Currenti & A. Bonaccorso.

First of all, I apologize for the delay in preparing the review. In this study, the authors apply an unsupervised machine learning approach (k-means algorithm) to borehole strainmeter signals (DRUV station) to automatize and enhance detection of strain variations associated with eruptive events. They focus their analyze on the intense activity of Etna volcano (> 60 lava fountains recorded) from December 2020 to February 2022.

The paper is well-written, the strain data pre-processing (i.e, correction of external influences) and the k-means approach are well-described. Based on the clustering approach, they are able to isolate common processes from strain signatures and further decipher the complexity of the volcano dynamics. Notably, the protocol helped to improve manual event detection (Carleo et al., 2022a) by revealing a new class of transient deformation on Etna volcano (Cluster 4). I only have minor comments :

1) It is interesting to see authors using strain rate in their analysis. Since, it is not often used in the literature, the authors may further comment on the interest of using strain rate for crustal deformation observation and how they complement the use of strain signals to study volcano dynamics.

2) I believe that the study is the very first to apply a machine learning approach to strainmeter data, and that is very interesting. Maybe the authors can insist on that aspect. This protocol is also promising to investigate other kind of processes using strain data (aseismic events, creep on volcanoes or faults), coseismic slip detection for instance. Maybe the authors can add a comment about that.

L. 303 : Agid. Geo In. → GeoHazards

---

## Referee Comment (RC2)

Clustering of eruptive events from high precision strain signals recorded during 2020-2022 lava fountains at Etna volcano (Italy) by L. Carleo, G. Currenti and A. Bonaccorso

My apologies for the day in making the review.
The manuscript shows an application of a machine learning technique for classification of lava fountain events at Mt Etna. To do so, they consider the period between December 2020-February 2022 when more than 60 lava fountain events took place. They cluster strain changes records in terms of features as the duration, the amplitude and the time derivative of the strain signal. The authors propose a novel approach to show the power of machine learning techniques to elude the subjective biases of manual classification of lava fountain episodes in terms of behavior. The manuscript shows an innovative method that could be very useful to an objective classification of eruption events. Nevertheless, the authors should improve the way the results are presented.

General comments

1. Introduction
The introduction is well structured. Some things to take into account that can be helpful to easily follow the work can be a very short description of the classification of eruption styles at Etna that is afterwards used at "Discussion and Conclusions" (TA, LF, LSLF, etc). This description can be introduced in paragraph that contain line #15.

2. Strain Changes during the Etna lava fountains in 2020-2022
The strain and strain rate signals are described in three parts. How is this description related to the change of curvature (inflection point) of the strain signal?

The use of 10 nstrain/ as the threshold to define the onset of period should be shortly explained at the text.

Figure 1b could be removed since there is not any reference to the figure at the main text of the manuscript.

Figure 2 (a,b) could be improved by adding shadows to the graphics to show up the different periods of the sequence of lava fountains. Furthermore, it could be helpful to print a vertical line on Figure 2(b) to highlight the strain event shown at Figure 2(c) .

3. Clustering of lava fountains
This section should be completely re-organized. A brief introduction about k-means method in terms of the way that the objects are grouped would be very welcome at the beginning of this section. Furthermore, I would explain main concepts such as centroid, points, etc as well as the drawbacks of the method.

Appendixes A and B are intended to explain the procedures developed to avoid the drawbacks of k-means algorithm and to show their quality. Nevertheless, the description of the results as well as the graphic results should have been removed from the appendixes and be thoroughly explained in the main text. The Table B1 with the features considered in the cluster analysis should be part of the main text too. Some attention should be paid to improve the way graphic results (Figure B1, Figure B2) are presented. Some ideas: the x-axis of Figure B1a and FigureB1b should be the same (both of them the parameter k which indicates the number of clusters). What is the meaning of elbow in Figure 1Ba? I am

not able to discern the average silhouette values for k = 4 mentioned in the main text. Figure B2 can be presented in a more compact way…

I think Table 3 should be Table 1.

4. Discussion

It is mentioned in the text (section 2) that 58 strain signals are used for the analyzed period. What is the ratio of these strain signals with respect to the total number of lava fountains? Is this an usual observational ratio for this kind of events? Is this ratio enough to describe the eruption different behaviors observed at Etna? How representative is the studied period in terms of the classification of the eruption behavior at Etna?
Although these aspects are briefly described in the first paragraph, some lines about quantification and significance of these aspects will be very welcome.

The classification is made with different features that are described in Table B1. Some comments about the final choice of the features to make the classification in terms of both the way the algorithm works and the relationship with the eruption styles and some of the possible physical processes involved will be very welcome. The time interval between the events is not included between the features considered to make the classification. Could you comment on that?

k-means algorithm allows the identification of cluster 4. In previous works, the events of cluster 4 are classified in different way. Could you briefly  comment on the impact of this fact? It should be done in terms of the power of the method for objective classification; in terms of eruption styles and in terms of the transition between Cluster 2 and Cluster 1 that Cluster 3 does not provide. Are the number of elements (Cluster 4 consists of 2 points ) enough to describe an eruption style? Any idea of what properties can modulate this behavior in terms of the eruption styles? Is there any relationship between the features that define the clusters and the characteristics of the eruption styles? In that case, what does it mean?

Figure 4 should include the temporal line showing the different periods (just the same as in Figure 2).

---

## Author Response (AR1)

ISTITUTO NAZIONALE DI GEOFISICA E VULCANOLOGIA

Catania, 3 April 2023

To: Giovanni Macedonio
Editor of NHESS

Dear Editor,

Please find herewith the revised version of the manuscript "*Clustering of eruptive events from high precision strain signals recorded during the 2020-2022 lava fountains at Etna volcano (Italy)*" by L. Carleo et al.

We are pleased that the reviewers and the Editor appreciated the manuscript and we are thankful for their positive comments and suggestions.

The manuscript has been modified, taking into account the suggestions of the editor and the two reviewers. A detailed point-by-point response to the recommendations provided by the editor and the reviewers is reported below. A "tracked-changes" version of the manuscript is also submitted.

We would like to thank the Editor and the reviewers for their constructive comments, which helped improve the manuscript.

Sincerely,
Gilda Currenti

**Reply to the Editor: « Clustering of eruptive events from high precision strain signals recorded during the 2020-2022 lava fountains at Etna volcano (Italy) » by L. Carleo, G. Currenti & A. Bonaccorso.**

The lines and figure numbers in this reply letter refer to the revised manuscript with tracked changes attached below.

*This paper shows an interesting and novel method for classification of lava fountains at Etna based on a machine-learning technique. The scientific approach is valid and potentially applicable to other volcanoes. I appreciate the suggestions of the reviewers because they help in improving the quality of the paper. In particular, I agree with reviewer #2 about the organization of section 3. Adding a brief introduction about the k-means method and a short descriptions of the used concepts (such as centroid, etc.) will simplify the reading.*

**RESPONSE**: We are grateful for the positive comments on the paper. As suggested by the editor and the reviewer #2 we added a new section 3 with the description of the algorithm and the procedure. This addition makes the paper clearer and more fluent.

*Technical comment: At page 10 of the manuscript with track-changes, attached to the reply of reviewer 2, at Line 188, "max{Sila,Ci,j} <= Sa,Ci". In the term on the right, probably do you mean "<= Sila,Ci"?*

**RESPONSE**: Thanks for spotting that. We corrected the typos.

**Reply to RC1: « Clustering of eruptive events from high precision strain signals recorded during the 2020-2022 lava fountains at Etna volcano (Italy) » by L. Carleo, G. Currenti & A. Bonaccorso.**

The lines and figure numbers in this reply letter refer to the revised manuscript with tracked changes attached below.

*First of all, I apologize for the delay in preparing the review. In this study, the authors apply an unsupervised machine learning approach (k-means algorithm) to borehole strainmeter signals (DRUV station) to automatize and enhance detection of strain variations associated with eruptive events. They focus their analyze on the intense activity of Etna volcano (> 60 lava fountains recorded) from December 2020 to February 2022.*
*The paper is well-written, the strain data pre-processing (i.e, correction of external influences) and the k-means approach are well-described. Based on the clustering approach, they are able to isolate common processes from strain signatures and further decipher the complexity of the volcano dynamics. Notably, the protocol helped to improve manual event detection (Carleo et al., 2022a) by revealing a new class of transient deformation on Etna volcano (Cluster 4). I only have minor comments.*

We thank the referee for the positive comments and the suggestions which helped to better highlight the results.

*1) It is interesting to see authors using strain rate in their analysis. Since, it is not often used in the literature, the authors may further comment on the interest of using strain rate for crustal deformation observation and how they complement the use of strain signals to study volcano dynamics.*
**RESPONSE**: As suggested by the referee, we better explain why we decided to use both strain and the strain rate signals to cluster the lava fountain events at Etna. We rewrite the period at Lines 111-133: "Such strain variations are the response to the decompression of the magmatic source feeding the lava fountain (Bonaccorso et al., 2013; Bonaccorso et al., 2016; Bonaccorso and Calvari, 2017; Bonaccorso et al., 2021; Currenti and Bonaccorso 2019). The time derivative of the filtered strain signal (strain rate signal), like other high-precision geodetic signals (Kozono et al, 2013; Ichihara, 2016), is expected to be related to the rate of magma chamber decompression and, thus to the speed of magma ascent (Hreinsdóttir et al., 2014). The Etna lava fountains grow gradually starting from Strombolian activity and evolving towards a sustained eruptive column. As already found in previous studies (Calvari et al., 2021; Calvari et al., 2022), the evolution of a lava fountain is well represented by both the strain and the strain rate signals. In Fig. 2c and 2d, the filtered strain and the filtered strain rate signals during the lava fountain on 1 – 2 July 2021 are shown as an example of the recorded variations. Typically, the strain and the strain rate signals show a sigmoid and a V shape, respectively. The different lava fountain phases can be described by dividing the signals in three main parts: in the initial part (Part I), when the Strombolian activity takes place, both the strain and the strain rate gradually decrease with time showing an elbow with a downward concavity; in the central part (Part II), the lava fountaining is persistent and the strain rate changes its slope abruptly reaching the absolute maximum value; in the final part (Part III), the eruptive activity starts declining and the strain rate inverts its trend reaching the pre-event level. To identify the beginning of the event, we focused on the strain rate signal. We first evaluated the amplitude of the background noise of the strain rate signal, $\sigma$, as the mean standard deviation in a moving time window of 3 hours. We found a value of $\sigma$ of 0.93 nstrain/h. The beginning of the variation $t_i$ was chosen concurrently with the time when the

beginning of the deformation rate can be clearly identified, namely when the strain rate exhibits a value of one order of magnitude higher than $\sigma$. Therefore, we selected $t_i$ as the time when the strain rate reaches -10 nstrain/h. The end of the variation $t_f$ was set when the sign of the strain rate becomes positive."

*2) I believe that the study is the very first to apply a machine learning approach to strainmeter data, and that is very interesting. Maybe the authors can insist on that aspect. This protocol is also promising to investigate other kind of processes using strain data (aseismic events, creep on volcanoes or faults), coseismic slip detection for instance. Maybe the authors can add a comment about that.*

**RESPONSE**: We thank the referee for the suggestion and modified the sentence at Line 57: "Clustering analyses on monitoring signals have already been performed in volcanology (Cirillo et al. 2022; Corradino et al., 2021; Langer et al., 2009; Nunnari, 2021; Romano et al., 2022; Unglert et al., 2016) but never applied on the strainmeter data for clustering eruptive events."

Moreover, we modified the sentences at Lines 277-279: "For the first time, an automated clustering analysis was applied on strainmeter data to provide an objective quantitative measure of similarities and differences between explosive eruptive episodes. In particular, we studied the lava fountain events that occurred at Etna in the period December 2020 - February 2022."

Following the referee's suggestion we added a comment on the use of the proposed protocol to investigate other kinds of deformation processes (Line 342): "The obtained results are very promising and encourage us to extend it to investigate other volcanic processes that engender strain changes such as magmatic recharges and intrusions.".

**Reply to RC2: « Clustering of eruptive events from high precision strain signals recorded during the 2020-2022 lava fountains at Etna volcano (Italy) » by L. Carleo, G. Currenti & A. Bonaccorso.**

The lines and figure numbers in this reply letter refer to the revised manuscript with tracked changes attached below.

*My apologies for the day in making the review.*
*The manuscript shows an application of a machine learning technique for classification of lava fountain events at Mt Etna. To do so, they consider the period between December 2020-February 2022 when more than 60 lava fountain events took place. They cluster strain changes records in terms of features as the duration, the amplitude and the time derivative of the strain signal. The authors propose a novel approach to show the power of machine learning techniques to elude the subjective biases of manual classification of lava fountain episodes in terms of behavior. The manuscript shows an innovative method that could be very useful to an objective classification of eruption events. Nevertheless, the authors should improve the way the results are presented.*

We thank the referee for the constructive criticism and suggestions that have been useful to improve the presentation of the results.

*1. Introduction*
*The introduction is well structured. Some things to take into account that can be helpful to easily follow the work can be a very short description of the classification of eruption styles at Etna that is afterwards used at "Discussion and Conclusions" (TA, LF, LSLF, etc). This description can be introduced in paragraph that contain line #15.*
**RESPONSE**: As yet, there is not a standard classification of explosive eruptive styles at Etna. Owing to the more frequent explosive eruptive events that occurred in the last decades, attempts to classify them have been made and the debate among experts is just open to reach a consensus. Adding a description in the Introduction of a specific classification results at Etna (Andronico et al. 2021) could be too restrictive and misleading. Therefore, we prefer to maintain this point in the discussion. However, this classification has been helpful to make a comparison and validate our results. Anyway, we agree that in the Introduction more focus on the event classification task is required to better follow the work. In this view, we rephrased and added new sentences at Lines 50-55: "In the recent past, attempts to classify the lava fountains at Etna have been made manually by the experts by comparing different geophysical and volcanological data. Andronico et al. (2021) manually found different eruptive styles at the Etna volcano on the basis of volcanological observations. Calvari et al. (2022) analysed three lava fountain episodes that occurred in 2021 with a multidisciplinary approach and gave insights into the different eruptive styles. Manual classification is time consuming since it involves a huge amount of data analysis and it is prone to subjective biases."

*2. Strain Changes during the Etna lava fountains in 2020-2022*
*The strain and strain rate signals are described in three parts. How is this description related to the change of curvature (inflection point) of the strain signal?*
**RESPONSE**: Also following the request of the first referee, we improved the description of the strain and strain rate signals by providing more detail about their evolution and the link with the eruptive activity. See Lines 111-133.

*The use of 10 nstrain/ as the threshold to define the onset of period should be shortly explained at the text.*

**RESPONSE**: Thanks for this request that allows us to describe in more detail the work performed to optimize the results. Indeed, the definition of the onset time of the lava fountain is challenging due to the gradual and also diverse evolutions for each event. We added the description of the choice of the threshold at Lines 127-133: "To identify the beginning of the event, we focused on the strain rate signal. We first evaluated the amplitude of the background noise of the strain rate signal, $\sigma$, as the mean standard deviation in a moving time window of 3 hours. We found a value of $\sigma$ of 0.93 nstrain/h. The beginning of the variation $t_i$ was chosen concurrently with the time when the beginning of the deformation rate can be clearly identified, namely when the strain rate exhibits a value of one order of magnitude higher than $\sigma$. Therefore, we selected $t_i$ as the time when the strain rate reaches -10 nstrain/h. The end of the variation $t_f$ was set when the sign of the strain rate becomes positive.".

*Figure 1b could be removed since there is not any reference to the figure at the main text of the manuscript.*

**RESPONSE**: We thank the reviewer for noting the lack of a reference to Fig. 1b. We added the reference at Line 26.

*Figure 2 (a,b) could be improved by adding shadows to the graphics to show up the different periods of the sequence of lava fountains. Furthermore, it could be helpful to print a vertical line on Figure 2(b) to highlight the strain event shown at Figure 2(c).*

**RESPONSE**: We preferred to not add too many elements in the Figure 2 to not cover the signal. We print an arrow in Fig. 2b in correspondence of the event shown in Fig. 2c-d.

*3. Clustering of lava fountains*
*This section should be completely re-organized. A brief introduction about k-means method in terms of the way that the objects are grouped would be very welcome at the beginning of this section. Furthermore, I would explain main concepts such as centroid, points, etc as well as the drawbacks of the method. Appendixes A and B are intended to explain the procedures developed to avoid the drawbacks of k-means algorithm and to show their quality. Nevertheless, the description of the results as well as the graphic results should have been removed from the appendixes and be thoroughly explained in the main text. The Table B1 with the features considered in the cluster analysis should be part of the main text too. Some attention should be paid to improve the way graphic results (Figure B1, Figure B2) are presented. Some ideas: the x-axis of Figure B1a and FigureB1b should be the same (both of them the parameter k which indicates the number of clusters). What is the meaning of elbow in Figure 1Ba? I am not able to discern the average silhouette values for k = 4 mentioned in the main text. Figure B2 can be presented in a more compact way…*
*I think Table 3 should be Table 1.*

**RESPONSE**: We introduced Section 3 entitled "Clustering the strain variations the k-means algorithm" that includes the methods and the procedure previously presented in Appendices A and B. The old Section 3 becomes the new Section 4 "Clustering results" whose contents were a bit re-organized. Also the Figures and the Table in the Appendices were moved in the main text.
We preferred to not change how the results were presented in Fig. 3. The x-axes of Fig. 3a and 3b do not represent the same quantity. Thanks to the reviewer's suggestion, we better explain in the main text the quantities reported in the plots to make it clear (see Lines 215-216 and Lines 222-224). This is now better specified also in the caption (Lines 227-230).

The elbow of Fig. 3a represents the most effective clustering solution in terms of $k$ and SSE. We added a sentence at the beginning of Section 3 where the Elbow method is presented.

The average Silhouette values indicated in the text (Lines 239-241) can be discerned from Fig. 3b since each value represents the average Silhouette value among the strain data points within the same cluster $i$-th. We better explain in Section 4 the results shown in the Fig. 3b.

As regards Fig. 4, we agree that the plots can be presented in a more compact way, for example by grouping Fig 4a,c,e and Fig 4b,d,f in only two figures. However, we prefer to present the plots separately in order to guarantee the legibility of all the figures since all the curves, especially the ones in Fig 4a,c,e, will strongly overlap making the evaluation of the curves difficult.

*4. Discussion*

*It is mentioned in the text (section 2) that 58 strain signals are used for the analyzed period. What is the ratio of these strain signals with respect to the total number of lava fountains? Is this an usual observational ratio for this kind of events? Is this ratio enough to describe the eruption different behaviors observed at Etna? How representative is the studied period in terms of the classification of the eruption behavior at Etna? Although these aspects are briefly described in the first paragraph, some lines about quantification and significance of these aspects will be very welcome.*

**RESPONSE**:

Due to the different eruptive activity there is not a full consensus on the classification and recognition of the eruptive events. In the analyzed period December 2020 - February 2022, the number of lava fountains recognized by different volcanologists may slightly vary because even for the experts themselves it is not simple to classify the events. Moreover, when the eruptive activity undergoes several phases of waning and waxing, often close-in-time events could be counted separately or as one (Calvari and Nunnari, 2022; Andronico et al., 2021). However, the discrepancy in the events counting is due to very few weak events.

Despite these differences among the experts, at most the total number of lava fountains in the analyzed period is 66 (Calvari and Nunnari 2022). We employed the protocol designed by Carleo et al. (2022b) to automatically detect the explosive events from the strain signal. By testing the protocol over a longer period from 20 November 2011 to 31 March 2021, Carleo et al. (2022b) showed that a true positive rate close to 1 could be achieved denoting that for each lava fountain a strain change is detected. Thanks to this high observational ratio, we can discern, select and study the signals recorded concurrently with almost all the different explosive events and use the strain signal as a further element together with other quantitative observations to assist the experts in classifying the events. By applying the protocol to the studied period, we detected 58 strain variations, all related to lava fountain events. Out of the 8 not recognized events, 2 were not recorded by the DRUV strainmeter because of malfunctioning of the station and the remaining 6 events are very weak or counted as sub-events.

During the lifetime of the DRUV station, the studied period is the most representative one in terms of frequency of occurrence and variety of the eruptive activity. Exceptionally, Etna experienced between February and April and between May and August on average 1 lava fountain per day (see Lines 80-81). For these reasons, this period is well adapted to conduct this study.

The Discussion has been improved adding the arguments suggested by the referee (see Lines 281-292).

*The classification is made with different features that are described in Table B1. Some comments about the final choice of the features to make the classification in terms of both the way the algorithm works and the relationship with the eruption styles and some of the possible physical processes*

*involved will be very welcome. The time interval between the events is not included between the features considered to make the classification. Could you comment on that?*

**RESPONSE**: The time interval between the event was not taken into account since we wanted to focus on the evolution of the event itself. Indeed, we had investigated the possible link between the event type and the inter-event time and we did not find a clear relationship. Just a fast inspection shows that the time interval between the events cannot be a distinctive feature of the eruptive episodes. For example, similar inter-event times are observed in the S1 and at the beginning of the S2 period, but the signal and the eruptive events were completely different belonging to Cluster 2-3 and Cluster 1, respectively.

*k-means algorithm allows the identification of cluster 4. In previous works, the events of cluster 4 are classified in different way. Could you briefly comment on the impact of this fact? It should be done in terms of the power of the method for objective classification; in terms of eruption styles and in terms of the transition between Cluster 2 and Cluster 1 that Cluster 3 does not provide. Are the number of elements (Cluster 4 consists of 2 points ) enough to describe an eruption style? Any idea of what properties can modulate this behavior in terms of the eruption styles? Is there any relationship between the features that define the clusters and the characteristics of the eruption styles? In that case, what does it mean?*

**RESPONSE**: At Etna, there is not a standard classification of eruptive styles. Attempts of classification have been made by Andronico et al. (2021) but the problem is still under debate. The use of automatic methods on geophysical signals for clustering the events can give more insights into the way lava fountain events can be classified. The analysis of the strain signal has demonstrated that it is a good proxy to objectively trace the lava fountain process. This quality makes the strain classification able to distinguish 4 clusters with diverse strength and duration. Such information has a strong impact in the alert system in place to manage volcanic crises. During lava fountain events which emit huge amounts of tephra in the atmosphere, knowledge on intensity and duration of the events has important implications, especially for civil aviation.

Supervised machine learning techniques are designed to work with huge amounts of data which need to be labeled to properly group them into clusters. Differently, unsupervised machine learning techniques, such as the k-means employed in this work, are developed to distinguish clusters of events even if they are formed by only a few elements. Therefore, we can consider the Cluster 4, formed by only 2 data points, convincing. Moreover, Cluster 4 identifies a group of events which are notably different from the others. The peculiarity of the events in Cluster 4 was also noticed in previous studies. Calvari and Nunnari (2022) analysed thermal camera images and estimated the duration of all the 2020-2022 episodes with three different approaches. A close inspection of their results show that the duration of the two events of Cluster 4 exhibits the largest values. Andronico et al. (2021) retrieved seismic parameters from volcanic tremor signals recorded during the eruptive episodes. The parameters related to the events of Cluster 4 show values higher than the average value estimated for the lava fountain events of the studied period.

The distinctive features of the clusters could be attributed to the degree of explosiveness and portion of effusive flows accompanying the event, that define the eruptive style. High precision geodetic signals (Kozono et al, 2013; Ichihara, 2016), and particularly the strain rate signal, are expected to be closely related to the speed of magma ascent (Hreinsdóttir et al., 2014) that is controlled by many parameters, such as exsolved and dissolved gas content, overpressure at depth, magma composition and conduit geometry of the plumbing system. To better understand the relationship between strain features and eruption style, a cross-validation with other observations (camera, seismic, satellite, geochemical and petrophysical data) is mandatory. This will be the next step of the research.

We thank the referee for the useful hints which help us to enrich the Discussion.

*Figure 4 should include the temporal line showing the different periods (just the same as in Figure 2).*

**RESPONSE**: We thank the reviewer for this suggestion and include the temporal line of Fig. 2 also in Fig. 6.